# Study of Stability, Cytotoxic and Antimicrobial Activity of Chios Mastic Gum Fractions (Neutral, Acidic) after Encapsulation in Liposomes

**DOI:** 10.3390/foods11030271

**Published:** 2022-01-20

**Authors:** Olga Gortzi, Magdalini Rovoli, Konstantinos Katsoulis, Konstantia Graikou, Despoina-Aikaterini Karagkini, Dimitrios Stagos, Dimitrios Kouretas, John Tsaknis, Ioanna Chinou

**Affiliations:** 1Department of Agriculture Crop Production and Rural Environment, University of Thessaly, 38446 Volos, Greece; 2Laboratory of Biochemistry, Faculty of Veterinary Medicine, University of Thessaly, 43100 Karditsa, Greece; mrovoli@uth.gr (M.R.); kkatsoulis@uth.gr (K.K.); 3Laboratory of Pharmacognosy & Chemistry of Natural Products, Department of Pharmacy, School of Health Sciences, National and Kapodistrian University of Athens, Panepistimiopolis-Zografou, 15771 Athens, Greece; kgraikou@pharm.uoa.gr (K.G.); ichinou@pharm.uoa.gr (I.C.); 4Department of Biochemistry and Biotechnology, University of Thessaly, Biopolis, 41110 Larissa, Greece; karagkin@uth.gr (D.-A.K.); stagkos@bio.uth.gr (D.S.); dkouret@uth.gr (D.K.); 5Department of Food Science and Technology, School of Food Science, University of West Attica, 12243 Egaleo, Greece; jtsaknis@uniwa.gr

**Keywords:** chios mastic gum, liposomes, encapsulation, stability, cytotoxic activity, antimicrobial activity

## Abstract

Mastic gum is a resinous sap produced by *Pistacia lentiscus* growing in the island of Chios (Greece) and has been recognized since Antiquity for its distinctive aroma as well as medical properties (antimicrobial, antioxidant, anti-inflammatory ones). The oral absorption of Chios Mastic gum (an insoluble polymer of poly-β-myrcene is among the most abundant contents) is poor due to its low water-solubility. We report in this study, two different Chios mastic gum extracts, the acidic mastic gum extract—AMGE—and the neutral one—NMGE, both prepared after removal of the contained polymer in order to ameliorate solubility and enhance in vivo activity. Liposomes are presented as a promising delivery system due to their physicochemical and biophysical properties to increase stability and absorption efficiency of the mastic gum extracts within the gastrointestinal (GI) tract. The aim of this study was to evaluate the stability in GI simulated conditions together with cytotoxic and antimicrobial activity of the two extracts (AMGE and NMGE) after encapsulation in a well characterized liposome formulation. Liposomes-AMGE complex showed an improved stability behavior in GI simulated conditions. Both assayed extracts showed significant dose dependent inhibition against the growth of liver cancer HepG2 cells and an interesting antimicrobial activity against several microorganisms. Conclusively, encapsulation could be evaluated as a beneficial procedure for further applications of mastic resin.

## 1. Introduction

Chios Mastic gum is a resinous sap produced from the tree *Pistacia lentiscus*, an evergreen tree belonging to the Anacardiaceae family, which is grown cultivated giving big amounts of its resin only on the Greek island of Chios. Chios mastic gum is used in the Mediterranean cuisine and has been recognized since ancient times both for its distinctive aroma and its healing properties (relief of upper abdominal discomfort, dyspepsia and peptic ulcer) based on its antibacterial, antifungal, antioxidant and anti-inflammatory activity [1,2,3,4,5,6,7,8].

The resin mainly appears to consist of organic ingredients, including a natural polymer (*cis*-1,4-poly-β-myrcene), volatile ingredients such as α-pinene, and myrcene that constitute the essential mastic oil, triterpenoids, phytosterols, polyphenolic molecules and a number of other bioactive secondary metabolites [7,9,10,11,12,13,14]. Mastic gum is highly insoluble in water, but is somewhat soluble in different organic solvents, including methanol, dimethylsulfoxide, acetone and chloroform. 

The oral absorption of crude resin due to *cis*-1,4-poly-β-myrcene is poor with very low water-solubility and reduces the bioavailability of the contained active compounds [6]. Therefore, mastic extracts could be prepared after removal of this insoluble polymer in order to ameliorate solubility and enhance potential in vivo activity. The use of suitable carriers, in order to overcome the drawbacks of such mastic gum extracts, could prove crucial.

Liposomes are small artificial vesicles of spherical shape that can be created from cholesterol and natural non-toxic phospholipids. Their properties differ considerably with lipid composition, surface charge, size, and the method of preparation [15]. Due to their hydrophobic and hydrophilic character, liposomes have an important ability to deliver drugs and other compounds to specific areas of an organism. Liposomes display flexible physicochemical and biophysical properties and therefore they are used as a delivery system for different delivery considerations [16]. These liposomes protect their contents by presenting a barrier, which is resistant to enzymes in alkaline solutions, digestive juices, bile salts, and intestinal flora that are generated in the human body. The contents of the liposomes are, therefore, protected from oxidation and degradation until they are delivered to the exact target gland, organ, or system where they will be utilized [17]. Liposomes encapsulation efficiency is not the most significant characteristic of these carriers but the stability and the absorption efficiency of them within the gastrointestinal tract (GI) is a major aspect, which leads to a growing scientific interest in understanding the digestion process of food colloids [18]. On the other hand, liposomes present therapeutic advantages as carriers of antimicrobial agents as they promote the efficiency of these agents due to the extensive interaction of the liposomes with the outer membrane of the bacterial cell, minimizing drug toxicity. They stimulate fusion with the cell membrane, enabling a high antibiotic delivery into the bacteria and potentially overcoming antibiotic resistance mechanisms. They also provide the time needed to allow antibiotics to realize their full bactericidal potential, leading to a reduction in the duration of antimicrobial agent’s treatment.

Cancer chemoprevention is the use of natural compounds acquired usually through diet or synthetic pharmacological agents to prevent, arrest, or reverse carcinogenesis at its earliest stages. Indeed, chemoprevention has been successfully achieved in animal experiments and clinical trials, and is considered one of the most important strategies for the fight against cancer [19]. Currently, two of the most important cancer types are liver and breast cancer. Liver cancer is the sixth most frequent cancer and the second leading cause of death from cancer in humans, while breast cancer is the most common invasive cancer in women [20,21]. Moreover, liver and breast cancer are considered among the most suitable types of malignancy for the application of chemoprevention [22,23]. In addition, many natural compounds have been used for cancer prevention [24,25,26]. The research on the cytotoxic effects of mastic is most recent [4,20,21,27], as the resin has been shown to exert antitumor growth activity through inhibition of cancer cell proliferation, angiogenesis, and inflammatory response. Therefore, studies have been performed to address the mechanisms of action of mastic at the genome-wide gene expression level [28]. In other studies, cytotoxic activity of solid and liquid types of mastic against fibroblasts and thirteen human tumor/leukemia cell types has been assessed [29], while they have further inhibited the spontaneous apoptosis of oral polymorphonuclear neutrophils [30]. These results led to the choice of evaluating the mastic extracts’ effect on the proliferation of liver and breast cancer cell lines.

Due to the well-known anti-helicobacter pylori and overall antimicrobial activity of mastic gum [2,7], mastic extracts and their liposomal encapsulated derivatives were evaluated for their antimicrobial activity against a panel of nine human pathogenic microbia (two Gram-positive, four Gram-negative and three fungi strains) and two food borne pathogens *Listeria monocytogenes* and *Salmonella poona* which are among the most important bacteria implicated in the food preservation. 

*Listeria monocytogenes* is a Gram-positive bacterium that can cause listeriosis, a very serious and often fatal disease in humans. Due to the severity of the disease as well as the capability of this pathogen to survive at refrigeration temperatures, it is very important for the food industry to find efficient agents for the control of *L. monocytogenes* [31]. *Salmonella* sp is the name of a genus of a rod-shaped, motile bacterium, non-spore forming, which belongs to Gram-negative and is the most frequently reported cause of food borne outbreaks of gastroenteritis worldwide. It can cause serious and sometimes fatal infections in some people, like children, elderly people and some with weakened immunity [32]. 

As a consequence, the aim of this study was to evaluate the stability of two Chios mastic gum extracts (AMGE and NMGE) after encapsulation in a well characterized liposome formulation in GI simulated conditions and the cytotoxic and antimicrobial activity of these liposomal complexes. 

## 2. Materials and Methods

### 2.1. Mastic Extraction (AMGE and NMGE)

Mastic gum used in this study was kindly supplied by the Chios Mastic Growers Association. Mastic gum (300 g) was diluted in ethyl acetate (300 mL) and then methanol (900 mL) was added. The mixture was left at room temperature for 3 days, and a layer of poly-β-myrcene was removed through filtration [10]. The total mastic extract was partitioned between aqueous 5% Na_2_CO_3_ (1 L) and ether (3.5 L). The organic phase was re-extracted three times with 5% Na_2_CO_3_ (1 L each time) and afforded the neutral fraction of mastic (NMGE, 95 gr). The aqueous phase was acidified with 1N HCl (3 L) and re-extracted with ether (6 L) and the organic phase afforded the acid fraction of mastic (AMGE, 120 g). 

Liposome Preparation- Ethanol Injection (EI): Liposomes were prepared by ethanol injection method after evaluation of three different preparation protocols (Thin-Film Evaporation (TFE), Freezing-Thawing (FT) and Ethanol Injection (EI)) by our laboratory, which have already been published [33]. Ethanol injection preparation protocol presented higher encapsulation efficiency and smaller mean particle size and polydispersity index compared to the other two protocols.

The method used was a modification of that reported by Dua et al. [34]. Briefly, a lipid phase consisting of a mixture of Phosphatidylcholine (PC) (10 mg/mL) (Sigma Aldrich, Taufkirchen, Germany), Cholesterol (CH) (2 mg/mL) (PC > 99% from egg yolk, CH > 90%, (Sigma Aldrich, Taufkirchen, Germany) and mastic gum extract (2 mg/mL) solution of ethanol is rapidly injected to a vast excess of aqueous phase (H_2_O d.d. + 0.3% NaCl) at 35 °C. The MLVs (Multi Lamellar Vehicles, are immediately formed. After MLV preparation, the dispersion was subjected to sonication (using a Misonix Sonicator S3000, Misonix Inc. Chicago, IL, USA) for 30 min in order to form SLV (Single Lamellar Vehicles) liposomes.

### 2.2. Stability Assessment of Liposomes for Gastrointestinal Stress 

Liposome’s stability was evaluated in simulated gastric juices (SGJ) and bile solution (BS) by turbidity measurements in predetermined time intervals (0 h, 3 h, 6 h and 24 h) by measuring the turbidity according to [35] at 450 nm using a turbidity meter (HI 83414, Hanna Instruments, Bedforshire, UK) at 25 °C. The SGJ and BS were prepared according to Pak et al. [36]. Briefly, SGJ were prepared by suspending pepsin from porcine stomach mucosa (Sigma, Taufkirchen, Germany) in sterile saline (0.5% *w*/*v*) to a final concentration of 0.22% *w*/*v*, adjusting pH to 2.0 with HCl. BS was prepared by dissolving porcine bile extract (Sigma, Taufkirchen, Germany) in DI water to a final concentration of 0.33%.

### 2.3. Cytotoxic Activity

XTT cell proliferation assay: The ability of the extracts to inhibit cancer cell growth was assessed using the XTT assay kit (Roche, Mannheim, Germany) as described previously [37]. The cells culture was in normal Dulbecco’s modified Eagle’s medium (DMEM, Gibco, Paisley, UK), which contains 10% (*v*/*v*) fetal bovine serum, 2 mM L-glutamine (Gibco, Paisley, UK), 100 units/mL of penicillin, and 100 units/mL of streptomycin (Gibco, Paisley, UK) in flasks appropriate for tissue culture at 37 °C in 5% CO_2_. 

Cell lines were subcultured into a 96-well plate with 1 × 10^4^ cells per well for HepG2 and MCF-7 cell in DMEM medium. After 24 h incubation, liver cancer HepG2 cells were treated with increasing concentrations (2–1000 μg/mL) of extracts in serum-free DMEM medium for 24 h. Thereafter, 50 μL of XTT-labeling reagent with 1 μL of electron coupling reagent was mixed to give 50 μL of XTT test solution, which was added to each well. The samples were incubated for 4 h and their absorbance was measured at 450 nm and 690 nm (reference wavelength) in a Bio-Tek ELx800 microplate reader (Winooski, VT, USA). Cells grown in DMEM serum-free medium were used as a negative control. In addition, the absorbance of each tested compound alone in DMEM serum-free medium and XTT test solution was tested at 450 nm. The absorbance values shown by the extracts alone were subtracted from those derived from cancer cell treatment with extracts. 

Data were calculated as a percentage of inhibition by the following formula: inhibition (%) = [(O.D._control_ − O.D._sample_)/O.D._control_] × 100
where O.D._control_ and O.D._sample_ indicated the optical density of the negative control and the tested substances, respectively. The concentration of extracts that caused 50% cellular proliferation inhibition (IC_50_) of cancer cells was calculated thereafter from the graph plotted percentage inhibition against extract concentration. The experiments were performed three times and on at least three separate occasions.

The extract exhibiting the strongest inhibitory activity against HepG2 cell growth after 24 incubation hours was selected for further examination. In particular, in order to examine the time dependent inhibitory activity, the extract was incubated with HepG2 cells at increasing concentrations (15–1000 μg/mL) for two more time-points, 48 and 72 h.

The extract exhibiting the strongest activity was also examined for inhibiting breast cancer MCF-7 cell growth at increasing concentrations (15–500 μg/mL) for 24, 48 and 72 h.

### 2.4. Statistical Analysis

All results are expressed as mean ± SD. For statistical analysis, one-way ANOVA was applied followed by Dunnett’s test for multiple pair-wise comparisons. Dose response relationships were examined by Spearman’s correlation analysis. Differences were considered significant at *p* < 0.05. All statistical analyses were performed with the SPSS software (version 14.0; SPSS) [37].

### 2.5. Antimicrobial Activity

The agar dilution method was used for the in vitro study of the sample’s antibacterial activity. The eleven tested microbia were two Gram-positive bacteria: *Staphylococcus aureus* (ATCC 25923), *Staphylococcus epidermidis* (ATCC 12228), four Gram-negative bacteria: *Escherichia coli* (ATCC 25922), *Enterobacter cloacae* (ATCC 13047), *Klebsiella pneumoniae* (ATCC 13883) and *Pseudomonas aeruginosa* (ATCC 227853), three human pathogen fungi: *Candida albicans* (ATCC 10231), *C. tropicalis* (ATCC 13801) and *C. glabrata* (ATCC 28838) and two food borne pathogens (*Listeria monocytogenes* (NCTC 11994) and *Salmonella poona* (NCTC 4840).

Stock solutions of the samples were prepared with absolute ethanol at 10 and 1 mg/mL, respectively. Serial dilutions of the stock solutions in broth medium (100 μL of Müller–Hinton broth or on Sabouraud broth) were prepared in a microtiter plate (96 wells). All tested organisms have a final cell concentration of 10^7^ cells/mL. In each well, 1 μL of the microbial suspension (the inoculum, in sterile distilled water) was added. The growth conditions and the sterility of the medium of each strain were controlled and then the plates were incubated. Minimum Inhibitory Concentration (MIC) was defined as the lowest concentrations preventing visible growth. Standard antibiotics (netilmicin and amoxicillin at concentrations of 4–88 μg/mL) were used for bacteria sensitivity checking. For the food borne pathogens netilmicin was used as control and for the human pathogen fungi 5-fluocytosine and amphotericin B (at concentrations 0.5–25 μg/mL). The experiments were carried out in triplicate, and the results were expressed as average values [38].

## 3. Results and Discussion

### 3.1. Stability Assessment of Liposomes for Gastrointestinal Stress

Liposomes-mastic gum extracts interaction was already studied and fully characterized by our laboratory [33] by using Fourier Transform Infrared (FT-IR) spectroscopy, surface morphology, Scanning Electron Microscopy (SEM), and size distribution using a particle size analyzer, Rancimat method and Differential Scanning Calorimetry (DSC). 

Liposomes together with the encapsulated substances can be absorbed by the GI epithelia. The mucus lining that coats and protects the GI epithelia sheds frequently, which as a result clears the substances that cannot penetrate the mucous layer quickly enough. Though there is a chance for liposomes to penetrate the mucous layer, the GI epithelia block the entrance of intact large liposomes into the circulation because they are commonly bigger in size in comparison with easily absorbable small molecules [39]. Hence, it is of great importance to study liposomes’ stability in simulated GI conditions.

Turbidity measurements (scattering particles) can be used as an alternative tool for the determination of liposomes’ stability [35]. In the case of liposomes, an increase in turbidity is interpreted as an increase in the aggregation of the vesicles due to the decrease in their stability [40]. Therefore, an increase in the percentage of optical density is tightly related to an increase in the turbidity and finally a decrease in the sample’s stability. The presented turbidity data in both SGJ and BS (Figure 1 and Figure 2) are based on the results of three different sets of experiments. Results show a higher stability of liposomes containing AMGE for both of aqueous media tested (SGJ and BS). 

Liposomes containing AMGE might be allocated and stabilized within the hydrophobic core of the liposomes at both the planar interface between lipid leaflets and within each acyl chain region due to the existence of a relatively more polar carboxyl group of AMGE, which easier interact with the polar head groups of phospholipids. Additionally, cholesterol in liposomes might help with the placement of AMGE in the phospholipid bilayer through its hydrophobicity [15,41]. The optical density (OD) measurements of all liposomes containing different forms of mastic in SGJ and in BS are presented in Figure 1 and Figure 2, respectively. 

The results in SGJ (Figure 1) showed that liposomes containing AMGE had an improved stability behavior in SGJ compared to liposomes containing NMGE or liposomes containing total mastic gum, respectively. This may be due to the ability of liposomes + AMGE to make a more stable complex in low gastric pH levels. 

The results in BS (Figure 2) showed that liposomes containing AMGE had an overall improved stability behavior in BS compared to liposomes containing NMGE or liposomes containing total mastic gum, respectively. However, the first 3 h of all three samples containing mastic gum extracts presented a similar stability behavior in simulated BS.

These results have shown that liposomes could act as an efficient protective barrier for mastic extracts and especially AMGE against gastrointestinal conditions. The well-organized assembly of phospholipids and cholesterol in liposomes could provide protection against membrane damage under low pH conditions.

### 3.2. Cytotoxic Activity

All tested extracts showed significant dose dependent inhibition against the growth of liver cancer HepG2 cells apart from the empty liposomes sample (Figure 3).

The inhibitory potency of the tested extracts against HepG2 cells was according to their IC_50_ values, liposomes containing NMGE (132 ± 8.5 μg/mL) > liposomes containing AMGE (255 ± 14.3 μg/mL) > liposomes containing total mastic gum (184 ± 12.3 μg/mL) (Table 1).

Subsequently, liposomes containing NMGE, the most potent among all tested extracts, was incubated for two more time-points, 48 and 72 h, with HepG2 cells. Dose-plot effects of liposomes containing NMGE at 48 and 72 h incubation time-points are shown in Figure 3. At these incubation time-points, liposomes containing NMGE’s IC_50_ values were much lower compared to 24 h, that is, they were 17 and 11 μg/mL at 48 and 72 h, respectively (Table 1).

Liposomes containing NMGE were also tested for its inhibitory activity against breast cancer MCF-7 cell growth after incubation for 24, 48 and 72 h. The extract’s dose-effect plots for this cancer cell line are shown in Figure 4. The IC_50_ values of liposomes containing NMGE, estimated from these dose-effect plots, were 18 ± 1.5, 12 ± 0.8 and 10 ± 0.6 μg/mL at 24, 48 and 72 h (Table 1). For comparison reasons, it is mentioned that the known drug sorafenib’s IC_50_ value was shown to be 7.5 μΜ (i.e., 3.5 μg/mL) against HepG2 cells, while doxorubicin’s IC_50_ value was shown to be 62 μg/mL against MCF-7 cells [42,43]. According to the American National Cancer Institute (NCI), an extract exhibited IC_50_ less than 30 μg/mL after 72 h incubation is characterized as cytotoxic against cancer cells, while if its IC_50_ is less than 20 μg/mL after 48 h incubation, then it is considered highly cytotoxic [44]. Thus, liposomes containing NMGE were shown high cytotoxic potency against both HepG2 and MCF-7 cells.

The IC_50_ values of the studied extracts against growth of cancer cells suggested their possible use for chemoprevention, since according to the literature an extract is considered to possess significant bioactivity, when it is active at concentrations up to 1000 μg/mL [45]. Moreover, according to the American Cancer Institute (NCI), the criterion of cytotoxic activity for a crude extract is of IC_50_ < 20 μg/mL after 72 h incubation [44].

### 3.3. Antimicrobial Activity

The two mastic extracts encapsulated in liposomes were assayed against eleven human pathogenic microbias, which belong to Gram-positive, Gram-negative and food borne bacteria as well as to fungi. 

According to the obtained results (Table 2, Table 3 and Table 4) the total mastic gum showed the strongest activity against all assayed microorganisms, followed by the acidic and the neutral mastic fraction, against the Gram positive (0.04–0.05 mg/mL) and negative strains (0.18–0.34 mg/mL) of human pathogenic, against the fungi (0.32–0.73 mg/mL) and foodborne bacteria (0.05–0.12 mg/mL). Regarding, the three fungi strains that were examined, the neutral mastic fraction had a slightly higher action compared to the acidic fraction.

Even though the two examined extracts exerted weaker antimicrobial activity compared to the total mastic extract (as expected), the results of the liposomes complexes were very promising (the acidic mastic fraction presents an overall higher antimicrobial activity in contrast to the neutral one).

Previous research has also shown that the stronger antimicrobial activity (against *Helicobacter pylori*) is demonstrated by the whole mastic extract followed by the acidic mastic fraction, while the neutral mastic fraction appeared completely inactive [46]. In another study, the antimicrobial effects of the essential oil from mastic resin have been investigated on food-borne organisms (*S aureus*, *Salmonella enteritidis*, *Lactobacillus plantarum*, *Pseudomonas* spp.) [3]. The authors reported strong antimicrobial effects, especially against the Gram positive bacteria (*S. aureus*, *L. plantarum*), while the Gram negative bacteria (*Pseudomonas* spp. and *S. enteritides*) were delayed from starting logarithmic growth [3]. A reduction in the MIC of liposomal ciprofloxacin and gentamicin, in comparison to the free drug, against most common resistant bacteria, such as *P. aeruginosa*, *K. pneumoniae* and *E. coli*, has been referred to in the bibliography [47]. According to the authors, in this study the improved antimicrobial activity was due to the efficient and extensive interaction of the liposomes with the outer membrane of the bacterial cell.

## 4. Conclusions

Mastic gum possesses various antioxidant and anti-inflammatory activities, which have been used widely through centuries. In this study, the applied use of an experimental liposomal formulation containing total mastic gum or two different mastic gum extracts (AMGE and NMGE) was evaluated regarding their stability in simulated gastric fluids and bile solution and their ability to act as a cytotoxic factor. These experiments have shown that the critical factors for mastic gum extract release rate proved to be the type and charge of the extract encapsulated in liposomes.

Regarding the stability assessment of liposomes encapsulated, total mastic and mastic extracts in GI simulated conditions, both in GSJ and BS, the complex of liposomes with AMGE showed an improved stability behavior compared to those that contained NMGE or total mastic gum. This indicates that liposomes + AMGE form a more stable complex compared to the other two examined mastic samples (total mastic gum and NMGE). The complex of AMGE + liposomes functions as a “control release carrier” and gives better stability properties. 

The inhibitory activity of the extracts against cell growth indicates that they potentially could be used further as potential chemoprotecting agents, with the NMGE extract being the most promising among the three studied mastic gum samples. Further studies are requested to clarify the mechanism by which liposomes + NMGE extracts interact with HepG2 cells (internalization or fusion with the cell membrane). 

Regarding the antimicrobial studies, the results revealed that the acidic mastic fraction exerted stronger antimicrobial activity against all assayed microorganisms.

Conclusively, results revealed that the acidic and neutral mastic fractions in liposome complex carriers presented an interesting cytotoxic and antimicrobial profile, which could be a very promising food and/or drug ingredient. Further in-depth analysis should be proposed and planned through a new series of further experiments to test and confirm this hypothesis towards future application.

## Figures and Tables

**Figure 1 foods-11-00271-f001:**
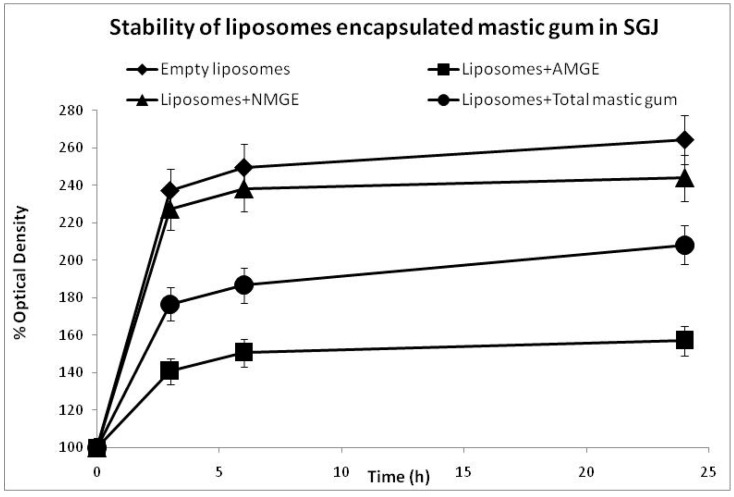
Stability of liposomes encapsulated mastic gum extracts in SGJ (simulated gastric juices). AMGE: acidic mastic gum extract; NMGE: neutral mastic gum extract.

**Figure 2 foods-11-00271-f002:**
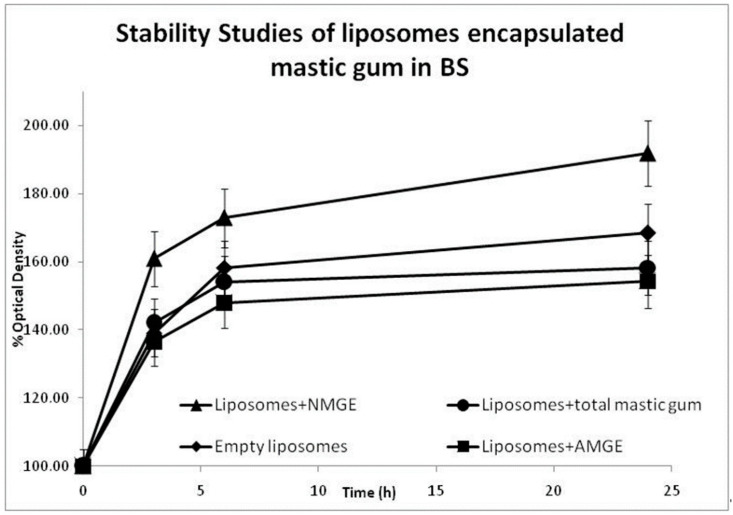
Stability of liposomes encapsulated mastic gum extracts in BS (bile solution). AMGE: acidic mastic gum extract; NMGE: neutral mastic gum extract.

**Figure 3 foods-11-00271-f003:**
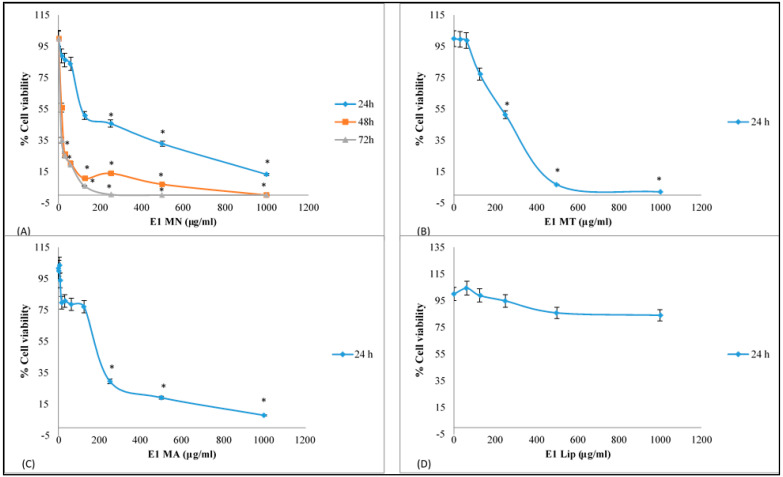
Dose–effect survival plots for mastic extracts against a human liver cancer HepG2 cell growth. (**A**) E1 MN: Liposomes contained NMGE, (**B**) E1 MT: Liposomes contained total mastic gum, (**C**) E1 MA: Liposomes contained AMGE, and (**D**) E1 Lip: Empty liposomes. Cell viability was estimated via XTT assay (each point represents mean ± sd of at least six replicate wells). * *p* < 0.05.

**Figure 4 foods-11-00271-f004:**
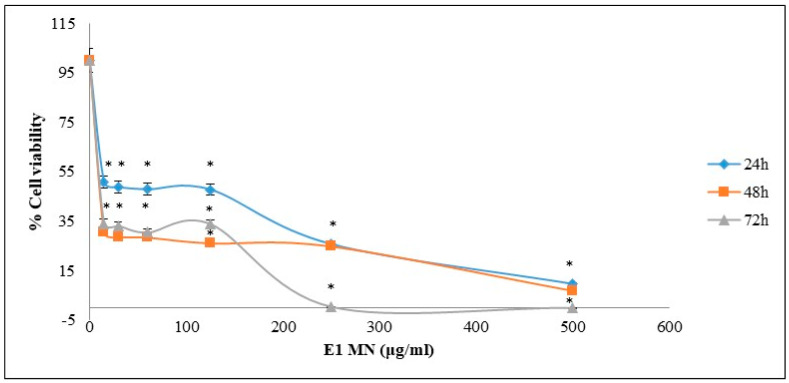
Dose–effect survival plots for liposomes containing NMGE against human breast cancer MCF-7 cell growth. Cell viability was estimated via XTT assay (each point represents mean ± sd of at least six replicate wells). * *p* < 0.05.

**Table 1 foods-11-00271-t001:** Evaluation of the inhibitory activity of the administered extracts against HepG2 and MCF-7 cancer cell growth expressed in IC_50_ values as determined based on XTT assay.

Compounds	IC50 Values μg/mL					
	HepG2			MCF-7		
	24 h	48 h ^b^	72 h ^b^	24 h ^b^	48 h ^b^	72 h ^b^
Liposomes contained NMGE	132 ± 6.7	17 ± 0.7	11 ± 0.9	18 ± 1.4	12 ± 1.1	10 ± 0.2
Liposomes contained AMGE	184 ± 20.2					
Liposomes contained total mastic gum	255 ± 17.9					
Empty liposomes	- ^a^					

^a^ Empty liposomes sample was not shown IC_50_ value at the tested concentrations. ^b^ at these time-points, cells were treated only with liposomes contained NMGE. AMGE: acidic mastic gum extract; NMGE: neutral mastic gum extract.

**Table 2 foods-11-00271-t002:** Antibacterial activity (inhibition zone and MIC mg/mL, respectively) of mastic extracts before and after liposome encapsulation.

Mastic Extract	Liposome Type	*S. aureus*	*S. epidermidis*	*P. aeruginosa*	*E. cloacae*	*K. pnaumioniae*	*E. coli*
Total mastic gum		22/0.04	20/0.05	17/0.21	15/0.25	13/0.34	18/0.18
Free AMGE		17/0.19	16/0.20	14/0.27	13/0.38	11/0.47	15/0.25
Free NMGE		14/0.45	13/0.52	12/0.73	11/0.88	10/1.0	12/0.64
Liposome + AMGE	EI	12/0.67	12/0.74	10/0.94	9/1.32	8/1.77	9/1.53
Liposome + NMGE	EI	10/1.38	10/1.50	9/1.71	9/1.94	8/1.98	8/1.94
Netilmicin		23/4 × 10^−3^	23/4 × 10^−3^	20/8.8 × 10^−3^	21/8 × 10^−3^	21/8 × 10^−3^	19/10 × 10^−3^
Amoxicillin		25/2 × 10^−3^	24/2 × 10^−3^	22/2.4 × 10^−3^	21/2.2 × 10^−3^	20/2.8 × 10^−3^	24/2 × 10^−3^

AMGE: acidic mastic gum extract; NMGE: neutral mastic gum extract.

**Table 3 foods-11-00271-t003:** Antifungal activity (inhibition zone and MIC mg/mL) of mastic extracts before and after liposome encapsulation.

Mastic Extract	Liposome Type	*Candida albicans*	*Candida tropicalis*	*Candida glabrata*
Total mastic gum		14/0.73	16/0.56	19/0.32
Free AMGE		10/1.10	11/0.98	13/0.78
Free NMGE		12/1.25	13/1.00	130.85
Liposome + AMGE	EI	8/1.90	9/1.88	10/1.81
Liposome + NMGE	EI	9/1.98	9/2.00	10/1.87
5-Flucytosine		21/1 × 10^−3^	22/1 × 10^−3^	25/0.5 × 10^−3^
AmphotericinΒ		22/1 × 10^−3^	23/0.5 × 10^−3^	23/0.4 × 10^−3^

AMGE: acidic mastic gum extract; NMGE: neutral mastic gum extract; EI: Ethanol Injection.

**Table 4 foods-11-00271-t004:** Activity against foodborne bacteria (inhibition zone and MIC mg/mL) of mastic extracts before and after liposome encapsulation.

Mastic Extract	Liposome Type	*Listeria monocytogenes*	*Salmonella poona*
Total mastic gum		21/0.12	25/0.05
Free AMGE		16/0.15	19/0.12
Free NMGE		13/0.27	15/0.20
Liposome + AMGE	EI	10/1.12	11/1.10
Liposome + NMGE	EI	10/1.27	10/1.24
Netilmicin		22/0.06	22/0.08

AMGE: acidic mastic gum extract; NMGE: neutral mastic gum extract; EI: Ethanol Injection.

## Data Availability

The datasets generated for this study are available on request to the corresponding author.

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
