# Peer review of "Study of Stability, Cytotoxic and Antimicrobial Activity of Chios Mastic Gum Fractions (Neutral, Acidic) after Encapsulation in Liposomes"

_foods, 2022, doi:10.3390/foods11030271_

Round 1
Reviewer 1 Report
- Authors either check liposome prepared on normal cell line or provide some evidencence that it is non toxic.
- In line no 66-67 sentences should be checked and corrected
- Has the authors used andy standard drug against the cell line. comparision needed.
- Different natural plants have been used against cancer prevention like Vincristine, vinblastine, artemesia (https://doi.org/10.1007/s12272-021-01328-4), an other need citation.
- Moreover the manuscript contain more than 30% palagriasim it need to be reduced to the acceptable limit some of sentence just copy Plagriasim report attached for referrences.
- reference for cytotoxicity also needed.
- Authors need to check grammatical error
Author Response
- In line no 66-67 sentences should be checked and corrected
The sentences have been corrected
- Has the authors used andy standard drug against the cell line. comparision needed.
It has not been used any standard drug in the cytotoxicity assays, however, known drugs’ cytotoxicity against HepG2 and MCF-7 cancer cell lines were found in literature and added in the manuscript for comparison reasons.
- Different natural plants have been used against cancer prevention like Vincristine, vinblastine, artemesia (https://doi.org/10.1007/s12272-021-01328-4), an other need citation.
As suggested more references for natural products exhibited cancer prevention activity were added in the manuscript [as: Bisht et al., 2021; Nagai et al., 2019; El-Sayed 2021; Katzenstein et al., 2021].
- Moreover the manuscript contain more than 30% palagriasim it need to be reduced to the acceptable limit some of sentence just copy Plagriasim report attached for referrences.
As several amendments and additional refs were added in the manuscript, the authors hope that the plagiarism issue has been overcome
- reference for cytotoxicity also needed.
According to Reviewer’s suggestion, reference for the assessment of the cytotoxic activity was added (Apostolou et al., 2013).
- Authors need to check grammatical error
The manuscript has been checked grammatically and for English spelling and appropriate changes/amendments were made accordingly
Reviewer 2 Report
Authors evaluated the stability in grastrointestinal tract simulated conditions of the two extracts, acid and neutral of Chios mastic gum , as wel as the antimicrobial and citotoxic activity, after the their encapsulation, with apropiate and innovative methods, and give the report very clear of them. They found good resultds and the conclussions are acoording to the experimental assays.
the therapeutics aplications of mastic resin seem promising.
Author Response
Thank you for your comments.
Reviewer 3 Report
Manuscript reference number: foods-1523155
Title: Study of stability, cytotoxic and antimicrobial activity of Chios mastic gum fractions (neutral, acidic) after encapsulation in liposomes
By: Olga Gortzi, Magdalini Rovoli, Konstantinos Katsoulis, Konstantia Graikou, Despoina-Aikaterini Karagkini, Dimitrios Stagos, Dimitrios Kouretas, John Tsaknis, Ioanna Chinou
General comments:
This article is an interesting contribution. But some sections of the manuscript should be reworked as suggested below to conform to journal standards. Particular attention must be paid to the extraction methods: currently it seems impossible to reproduce the preparation of the extracts. In addition, a statistical analysis seems to have been performed (see lines 176-177) but in tables no result was expressed as mean ± SD and in the graphs no error-bar was shown. In general, English should be improved and some sentences should be clarified.
Minor and major concerns:
- Lines 46/51: the ‘cis’ stereodescriptor should be written in italics
- Lines 111-118: extraction methods should be more precise as indicated:
A quantity of mastic gum (masse used) was diluted in ethyl acetate (volume) and then methanol (volume) was added. The mixture was let stand at room temperature (how long?) and a layer of poly-β-myrcene was decanted. The clear supernatant solution was obtained by filtration. The total mastic extract was partitioned between aqueous 5% Na2CO3 (volume) and ether (volume). The organic phase was re-extracted three times with 5% Na2CO3 (volume) and afforded the neutral fraction of mastic (NMGE, quantity recovered). The aqueous phase was acidified with 1N HCl (volume or final pH of the solution) and re-extracted with ether (volume) and the organic phase afforded the acid fraction of mastic (AMGE, quantity recovered).
- Line129 : Abbreviations should be explained the first time they appear. MLV should be defined at line 129 and not 130.
- Line 192 : the authors should precise the solvent used to prepare stock solutions
- Lines 260-262: the authors should name the different parts of the figure and explain in the figure 3 title to which liposomes (NMGE or AMGE or total or empty) the curves correspond.
- Lines 264/267/head of column of table 1/275/280/288/292: the authors should write ‘50’ in subscript when discussing IC50
- Line 298: a word seems to be missing “according to the obtained ?”
- Lines 303/306/309: units of the inhibition zone should be added
- In table 3: 5-flucytosine should be corrected by fluorocytosine
- Doi are missing in some references: 6 (1055/s-2006-960856), 7 (10.2174/092986712800229014), 30 (10.4172/2157-7110.1000355), 36 (10.1007/s11745-001-0846-x), 39 (10.9734/EJMP/2012/1040)
- Reference 38 should be completed: journal, volume, doi
- English has to be improved: For exemple at line 74 an extra “of” is written, line 69-70 or 86-87 sentences should be enhanced, line 91 a l is missing in ‘wellknown’, line 192 remove the extra ‘r’ in both medium, line 200 a r is missing in fluorocytosine, line 278 ‘was’ should be corrected in ‘were also tested’.
Author Response
This article is an interesting contribution. But some sections of the manuscript should be reworked as suggested below to conform to journal standards.
- Particular attention must be paid to the extraction methods: currently it seems impossible to reproduce the preparation of the extracts.
The whole phytochemical method has been updated and all requested details have been added
- In addition, a statistical analysis seems to have been performed (see lines 176-177) but in tables no result was expressed as mean ± SD and in the graphs no error-bar was shown.
Thank for your comment. By omission, SD values and error bars were missed. So, as suggested, SD values were added in Tables. Error bars were also added in the liposomes stability graphs in Figure 1 and 2.
- In general, English should be improved and some sentences should be clarified.
The manuscript has been checked grammatically and for English spelling and all appropriate amendments were introduced
Minor and major concerns:
- Lines 46/51: the ‘cis’ stereodescriptor should be written in italics
The comment was followed with appropriate correction
- Lines 111-118: extraction methods should be more precise as indicated:A quantity of mastic gum (masse used) was diluted in ethyl acetate (volume) and then methanol (volume) was added. The mixture was let stand at room temperature (how long?) and a layer of poly-β-myrcene was decanted. The clear supernatant solution was obtained by filtration. The total mastic extract was partitioned between aqueous 5% Na2CO3 (volume) and ether (volume). The organic phase was re-extracted three times with 5% Na2CO3 (volume) and afforded the neutral fraction of mastic (NMGE, quantity recovered). The aqueous phase was acidified with 1N HCl (volume or final pH of the solution) and re-extracted with ether (volume) and the organic phase afforded the acid fraction of mastic (AMGE, quantity recovered).
The following details have been added “Mastic gum used in this study was kindly supplied by the Chios Mastic Growers Association. A quantity of mastic gum (300g) was diluted in ethyl acetate (300mL) and then methanol (900mL) was added. The mixture was let stand at room temperature (3 days) and a layer of poly-β-myrcene was decanted. The clear supernatant solution was obtained by filtration. The total mastic extract was partitioned between aqueous 5% Na2CO3 (1L) and ether (3.5 L). The organic phase was re-extracted three times with 5% Na2CO3 (1L each time) and afforded the neutral fraction of mastic (NMGE, 95 gr). The aqueous phase was acidified with 1N HCl (3L) and re-extracted with ether (6L) and the organic phase afforded the acid fraction of mastic (AMGE, 120g). “
- Line129 : Abbreviations should be explained the first time they appear. MLV should be defined at line 129 and not 130.
MLV abbreviation was defined at line 129
- Line 192 : the authors should precise the solvent used to prepare stock solutions
The solvent has been clarified in the text (Methods). Absolute ethanol has been used as solvent to prepare stock solutions. Blind control of the solvent used has been also assayed, not showing antimicrobial activities.
- Lines 260-262: the authors should name the different parts of the figure and explain in the figure 3 title to which liposomes (NMGE or AMGE or total or empty) the curves correspond.
The different parts of Fig. 3 were named and in the title, it was explained which extract corresponds to which curve.
- Lines 264/267/head of column of table 1/275/280/288/292: the authors should write ‘50’ in subscript when discussing IC50
As suggested by the Reviewers, in all manuscript, the ‘50’ was written as subscript in ‘IC50’.
- Line 298: a word seems to be missing “according to the obtained?”
Following the suggestion. The sentence has been completed “According to the obtained results …”
- Lines 303/306/309: units of the inhibition zone should be added
The units of the inhibition zone have been added to the text “...against the Gram positive (0.04-0.05 mg/mL) and negative strains (0.18-0.34 mg/mL) of human pathogenic, against the fungi (0.32-0.73 mg/mL) and foodborne bacteria (0.05-0.12mg/mL).”
- In table 3: 5-flucytosine should be corrected by fluorocytosine
Flucytosine, also known as 5-fluorocytosine (5-FC), they are synonyms and flucytosine is fully approved and correct. The authors should prefer to stay with flucytosine, as used in many previous publications, indeed
- Doi are missing in some references: 6 (1055/s-2006-960856), 7 (2174/092986712800229014), 30 (10.4172/2157-7110.1000355), 36 (10.1007/s11745-001-0846-x), 39 (10.9734/EJMP/2012/1040)
All missing doi have been added in the List of references
- Reference 38 should be completed: journal, volume, doi
All the details regarding the ref have been added.
- English has to be improved: For exemple at line 74 an extra “of” is written, line 69-70 or 86-87 sentences should be enhanced, line 91 a l is missing in ‘wellknown’, line 192 remove the extra ‘r’ in both medium, line 200 a r is missing in fluorocytosine, line 278 ‘was’ should be corrected in ‘were also tested’.
The whole manuscript has been checked grammatically and for English spelling. Dealing with the suggestion to use the fluorocytocine instead of flucytocine, as explained before they are synonyms, both correct and the authors have a preference to keep a consistency with previous publications and keep “flucytocine”
Reviewer 4 Report
The aim of this study was to evaluate the stability in gastrointestinal simulated conditions, the cytotoxic and antimicrobial activities of two different Chios mastic gum fractions (the acidic and neutral fractions) after encapsulation in a well characterized liposome formulation.
This research is important and can bring valuable information with practical application.
The presented research is well-planned and the manuscript is generally well organized. There were used an appropriate experimental design.
Therefore, the work could be of interest but some points have to be considered:
- The Introduction provides some data on the stage knowledge of this issue. In the introduction, the research design is not clear presented, with a clear presentation of the steps and methods used for answering the research question.
- Some information regarding liposome entrapment efficiency (in order to know the degree of liposomal extracts entrapment) must be included, and the discussions could also include some correlations between their active components and biological effects.
- The discussions interpret the findings in view of the results obtained in this research. But, the discussions could take in attention more other literature data related with this subject.
- References could be complete.
- It should be improved English language.
Author Response
Therefore, the work could be of interest but some points have to be considered:
- The Introduction provides some data on the stage knowledge of this issue. In the introduction, the research design is not clear presented, with a clear presentation of the steps and methods used for answering the research question.
In the Introduction a paragraph and some more refs were added in order to support the scope of the present study as well as the strength of the obtained results.
- Some information regarding liposome entrapment efficiency (in order to know the degree of liposomal extracts entrapment) must be included, and the discussions could also include some correlations between their active components and biological effects.
The entrapment efficiencies of mastic extracts were not determined. In each case mastic compounds entrapped in liposomes represents complexes with new physicochemical properties and their biological activity was examined in the present work.
- The discussions interpret the findings in view of the results obtained in this research. But, the discussions could take in attention more other literature data related with this subject.
To our knowledge it is the first time that Chios Mastic extracts are trapped in liposomes. However, research results obtained for antimicrobial and cytotoxic ability of mastic extracts from the recent literature, have been added to the manuscript for comparative evaluation.
- References could be complete.
All the details regarding the references have been added.
- It should be improved English language.
Τhe whole manuscript has been checked grammatically and for English spelling
Round 2
Reviewer 3 Report
Thank you for detailing procedures. No more concern to add.
Reviewer 4 Report
In accordance with the reviewers' suggestions, the changes made by the authors bring clarifications and improve the quality of the manuscript.